# Do NOT Think That Much for 2+3=? On the Overthinking of Long Reasoning Models

**Xingyu Chen** [* 1 2]   **Jiahao Xu** [* 2]   **Tian Liang** [* 2]   **Zhiwei He** [* 1 2]   **Jianhui Pang** [2]   **Dian Yu** [2]   **Linfeng Song** [2]
**Qiuzhi Liu** [2]   **Mengfei Zhou** [2]   **Zhuosheng Zhang** [2]   **Rui Wang** [1]   **Zhaopeng Tu** [2]   **Haitao Mi** [2]   **Dong Yu** [2]

## Abstract

The remarkable performance of long reasoning models can be attributed to their ability to emulate human-like long-time thinking during inference. These models employ extended chain-of-thought (CoT) processes, exploring multiple strategies to enhance problem-solving capabilities. However, a critical question remains: *How to intelligently and efficiently scale computational resources during testing*. This paper presents the first comprehensive study on the prevalent issue of **overthinking** in these models, where long reasoning models generate redundant solutions that contribute minimally to accuracy and diversity, thereby wasting computational resources on simple problems with minimal benefit. We introduce novel efficiency metrics from both outcome and process perspectives to evaluate the rational use of computational resources by long reasoning models. Using a self-training paradigm, we propose strategies to mitigate overthinking, simplifying reasoning processes without compromising accuracy. Experimental results show that our approach successfully reduces computational overhead while preserving model performance across a range of testsets with varying difficulty levels, such as GSM8K, MATH500, GPQA, and AIME. Our code is available at https://github.com/galaxyChen/overthinking.

## 1. Introduction

The OpenAI o1 model (OpenAI, 2024) and its replicas (Qwen, 2024; Guo et al., 2025; Kimi et al., 2025; DeepSeek, 2025) exemplify the state-of-the-art in AI rea-

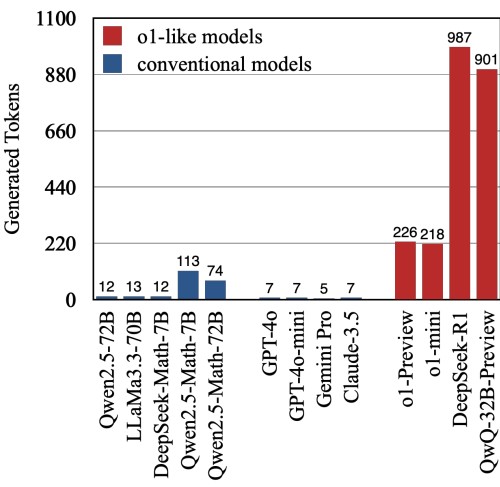

*Figure 1.* Illustration of **overthinking issue**: long reasoning models (in red color) spend much more tokens than conventional LLMs (in blue color) on a simple question "What is the answer of 2 plus 3?".

soning. Their success is largely attributed to mimicking human-like long-term thinking before responding to a question. Specifically, long reasoning models cultivate a long chain of thoughts (CoT), explore multiple strategies, break down complex steps, and perform double-checking, which ultimately enhance their ability to tackle intricate reasoning tasks. This approach, known as "scaling test-time compute", involves allocating more computational resources during the model's inference phase to generally yield more accurate responses.

While effective, a critical yet underexplored question remains: **Are we scaling test-time compute efficiently and intelligently?** This study provides an initial exploration of this problem. We first observe that long reasoning models exhibit significant **overthinking** issues. Specifically, they tend to expend excessive compute (in terms of tokens or thinking rounds) on questions that are exceptionally simple or for which the answer is already evident. For example, Figure 1 compares the token usage of long reasoning models with conventional models when answering the question, "what is the answer of 2 plus 3?" On average, long reasoning

---
[*]Equal contribution   [1]Shanghai Jiao Tong University, China   [2]Tencent.   Correspondence to:   Zhaopeng Tu <zptu@tencent.com>, Rui Wang <wangrui12@sjtu.edu.cn>.

*Proceedings of the 42nd International Conference on Machine Learning*, Vancouver, Canada. PMLR 267, 2025. Copyright 2025 by the author(s).

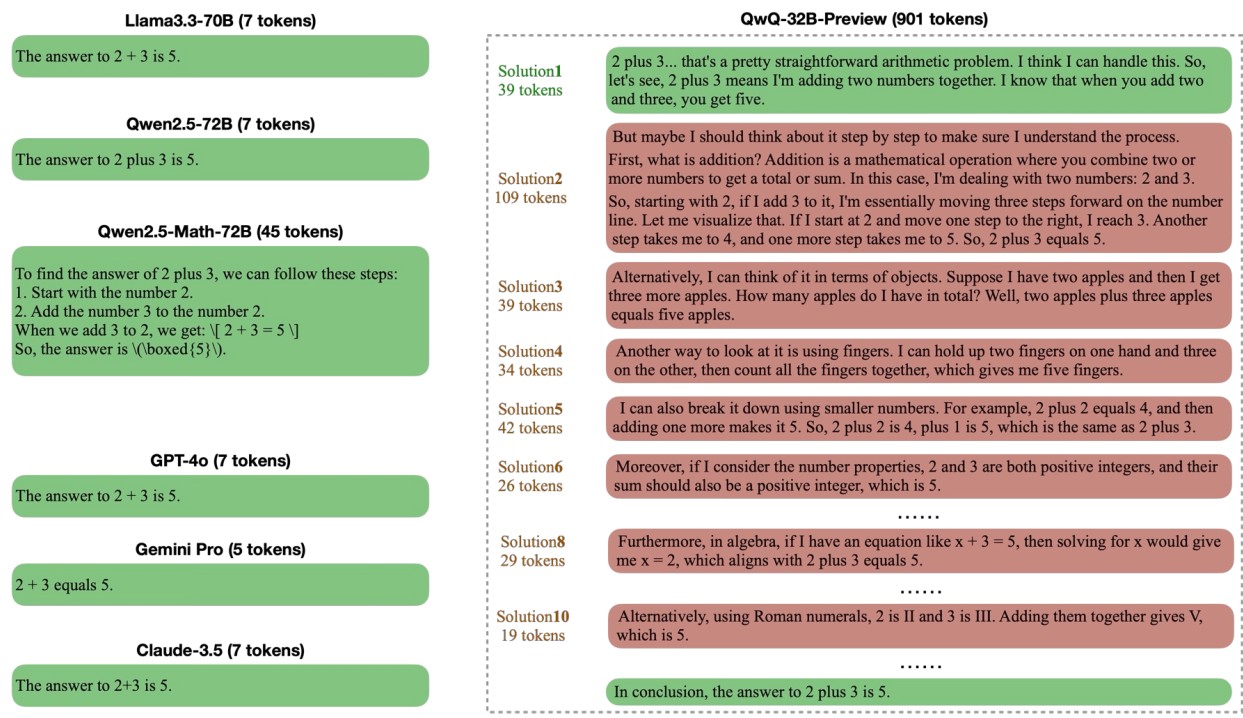

*Figure 2.* An example of overthinking issue for QwQ-32B-Preview model's output response that consists of 13 solutions. Each colored box contains a solution with the same explicit answer 5. The model repeatedly produces solutions with identical answers, demonstrating redundancy and a lack of diversity. We also list the outputs of other conventional LLMs for reference.

models consumed about **20x** more tokens than conventional models to reach the same answer. Figure 2 illustrates a concrete example where excessive long thinking results in generating 13 solutions for this trivially simple question. Across extensive analyses of mathematical benchmarks, we found these overthinking patterns: (1) contribute minimally to improving accuracy, (2) lack diversity in reasoning strategies, and (3) occur more frequently with simple problems.

The overthinking observed in long reasoning models reveals inefficiency in inference and highlights fundamental limitations in their reasoning and decision-making processes. We assert that reasoning involves not only accuracy but also the application of the appropriate level of complexity based on the problem's requirements. This insight motivates our exploration of studying and mitigating overthinking. To address this, we propose two metrics from both outcome and process perspectives to evaluate long reasoning models' efficiency. These metrics help provide a comprehensive assessment of the **efficiency** of long reasoning models, augmenting the commonly-used **effectiveness** metrics.

To mitigate overthinking without introducing external information, we adopt a self-training paradigm that teaches the model to produce more concise responses using its own generated outputs. We simplify the generated responses by removing redundant solutions while maintaining basic

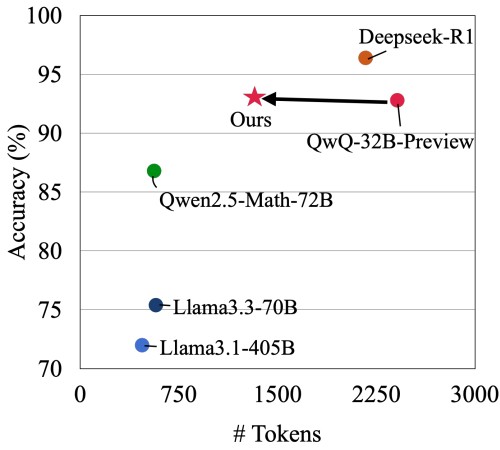

*Figure 3.* Token-accuracy plot on MATH500. Our method reduces the overthinking issue when applied to QwQ-32B-Preview.

reflexivity. Experimental results across testsets of varying difficulty levels (e.g., ASDIV, GSM8K, MATH500, GPQA, and AIME) demonstrate our approach's effectiveness and robustness in mitigating overthinking issues. For instance, as shown in Figure 3, our approach can reduce token output by 44.5% while maintaining accuracy on the widely-used MATH500 testset as applied to QwQ-32B-Preview.

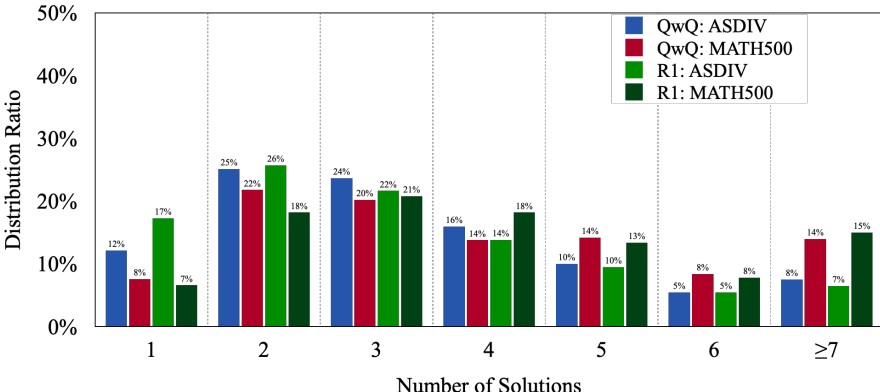

*Figure 4.* Distribution of solution counts in generated responses for different test sets and models (QwQ-32B-Preview ("QwQ") and DeepSeek-R1 ("R1")).

In summary, our contributions are three-fold:

1. We present the first study offering both a definitive explanation and comprehensive analysis of the overthinking issue, showing that long reasoning models often expend unnecessary computational resources on redundant solutions that contribute minimally to final outcomes.

2. We introduce metrics considering both outcome and process aspects to assess the efficiency of long reasoning models.

3. We explore several strategies to mitigate overthinking, significantly reducing generated tokens while maintaining model performance across testsets of varying difficulty.

## 2. Observing Overthinking Issues

In this section, we present a comprehensive analysis of outputs generated by long reasoning models. First, we provide a basic illustration of the solution distribution in responses from these models (§ 2.1). We then identify two inefficiencies in long reasoning responses: their limited contribution to accuracy (§ 2.2) and diversity (§ 2.3). To evaluate these inefficiencies empirically, we propose two efficiency metrics based on our observations. Finally, we present empirical results in § 2.4 and conclude that *long reasoning models often overthink, particularly with easier math problems.*

### 2.1. Solution Distribution of long reasoning Models

**Experimental Setup**   We use two test sets, including AS-DIV (Miao et al., 2020) and MATH500 (Hendrycks et al., 2021). ASDIV is an elementary-level math test set, while MATH500 is a high school-level math test set. We mainly investigate two widely recognized long reasoning models featuring a visible thinking process: Qwen-QwQ-32B-Preview (Qwen, 2024) and DeepSeek-R1 (DeepSeek, 2025).

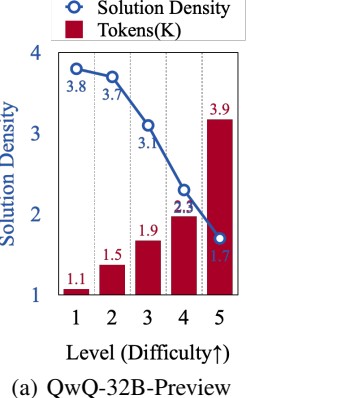
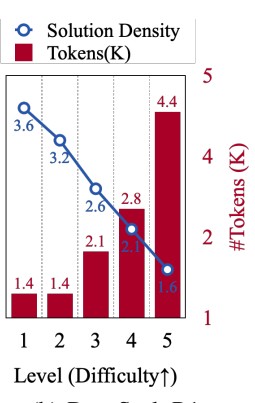

(a) QwQ-32B-Preview          (b) DeepSeek-R1

*Figure 5.* Average solution density (the number of solutions per 1000 tokens) and the number of tokens in generated responses across different difficulty levels of the MATH500 test set.

**Solution Distribution**   In this paper, we define *solution* as part of the full model generation that contains an answer explicitly. For example, in Figure 2, each solution in the QwQ-32B-Preview generation contains the answer 5. We use the Llama-3.3-70B model to separate solutions from generated responses. Figure 4 shows the distribution of solutions in generated responses. Generally, long reasoning models produce 2 to 4 solution rounds for most instances, covering 56% to 65% of cases for QwQ-32B-Preview across the test sets and 57% to 62% for DeepSeek-R1. Regarding different test sets, long reasoning models tend to generate more solutions on average for easier test sets. Since these models naturally produce more tokens for harder problems, we define *Solution Density* as the number of solutions per 1,000 tokens to compare the extent of overthinking across different difficulty levels. The solution density of QwQ-32B-Preview is 5.5 on ASDIV and 2.7 on the most difficult MATH500 test set. Similarly, DeepSeek-R1 achieves solu-

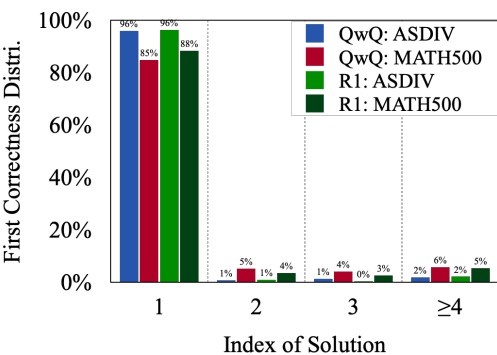

*Figure 6.* Distribution of occurrences for the first correct answer. Later solutions marginally contribute to improvements in accuracy

tion densities of 4.2 and 2.4, respectively. As shown, both QwQ-32B-Preview and DeepSeek-R1 exhibit more severe overthinking in simpler problems.

To further empirically validate this finding, we conducted an analysis across various difficulty levels in the MATH500 test set, as illustrated in Figure 5. Both QwQ-32B-Preview and R1 generate more tokens for difficult problems, and their solution density decreases as question difficulty increases. For example, QwQ-32B-Preview achieves a solution density of 3.8 for the simplest (level 1) questions, indicating that, on average, every 1,000 tokens contain more than three solutions, whereas the solution density for level 5 problems drops to 1.7. Although R1 performs slightly better, it still produces multiple solutions for simple questions. These results support our claim that **long-reasoning models tend to overthink simpler problems**.

### 2.2. Efficiency on Accuracy Improvements

**Intuition** In the example in Figure 2, we observe that the initial round of solutions already yields the correct answer. Subsequent solutions, which account for the majority of generated tokens, do not enhance accuracy. Based on this observation, we empirically investigate whether later solutions contribute to accuracy improvements. Specifically, for all cases where long reasoning models produce the correct answer in the response, we calculate the distribution of occurrences for the first correct answer, termed the "first correctness distribution". If more correct answers appear in earlier solutions, then the subsequent solutions contribute minimally to accuracy improvement, indicating reduced efficiency.

**Observation** Figure 6 illustrates the first correctness distribution across the test sets and models. In more than 80% of cases, the initial round of solutions produces the correct answer. Notably, the first round generally comprises less than 60% of the total tokens generated, suggesting that the

extended CoT might not significantly enhance accuracy. For instance, the average length of the first round of solutions for QwQ-32B-Preview on the ASDIV test set is 287 tokens, constituting only 38.7% of the entire response. These results suggest that **later solutions marginally contribute to improvements in accuracy**.

**Outcome Efficiency Metric** Based on the above observation, we propose an outcome efficiency metric to empirically evaluate how effectively later solutions contribute to accuracy improvements. The outcome efficiency metric, denoted $\xi_O$, is defined by the following formula:

$$\xi_O = \frac{1}{N} \sum_{i=1}^{N} \sigma_i \frac{\hat{T}_i}{T_i} \tag{1}$$

where $N$ is the number of instances in a given test set. $\sigma_i$ denotes whether the evaluated model can produce a correct answer in the response:

$$\sigma_i = \begin{cases} 1, & \text{if at least one solution in response is correct} \\ 0, & \text{otherwise} \end{cases}$$

$T_i$ is the total number of tokens produced for the $i$-th instance, and $\hat{T}_i$ denotes the efficient tokens that contribute to reaching the correct answer:

$$\hat{T}_i = \begin{cases} \text{\#tokens to first arrive at correct answer,} & \sigma_i = 1 \\ T_i, & \sigma_i = 0 \end{cases}$$

Intuitively, if a model correctly answers at an early stage, the tokens generated thereafter do not contribute to improving accuracy and are considered inefficient. Consider Figure 2 as an example: The first solution correctly addresses the problem with $\hat{T} = 39$. Consequently, $\xi_O = \frac{39}{901} = 4.3\%$, which can be considered extremely inefficient.

### 2.3. Efficiency on Diverse Thinking

**Intuition** Some researchers might argue that while solving an easy math problem may appear straightforward, approaching it from different perspectives can deepen understanding and build flexibility in mathematical thinking, which is also valuable. Consider the example output of QwQ-32B-Preview in Figure 2: Solution 1 states the basic fact that 2 plus 3 equals 5; Solution 2 breaks the addition into smaller steps; Solution 3 uses a counting objects analogy. These three solutions provide different reasoning strategies. However, Solution 4 repeats Solution 3, and Solution 5 repeats Solution 2 using similar reasoning strategies. In this section, we empirically examine the diversity among solutions within a response.

**Observation** To empirically evaluate whether later solutions provide new reasoning strategies, we introduce the

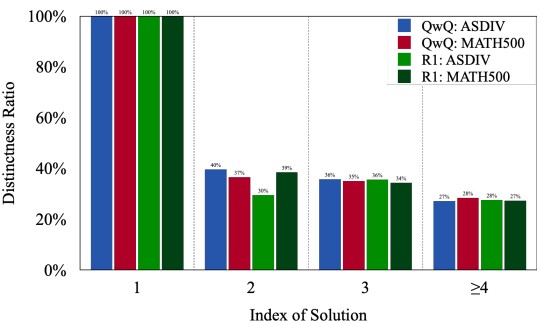

*Figure 7.* The ratio of diverse solutions at each solution index. Later solutions tend to repeat earlier ones.

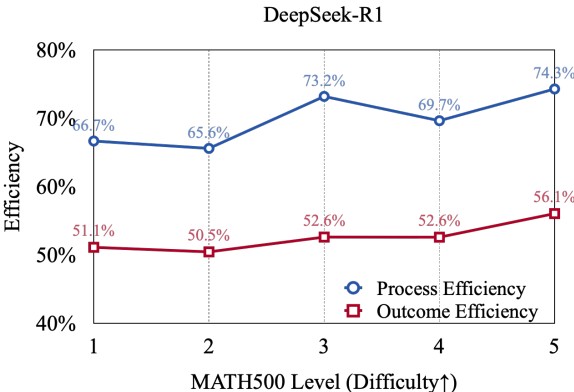

*Figure 8.* Efficiency results across different levels of MATH500.

"distinctness ratio" as the measure for the ratio of distinct solutions for each data index. Suppose the response to the $i$-th problem from a given test set can be divided into $M_i$ solutions. We denote the set of solutions as $R_i = \{s_i^1, \ldots, s_i^m, \ldots, s_i^{M_i}\}$. Let $S^m = \{s_1^m, \ldots, s_k^m, \ldots, s_K^m\}$ be the set of $m$-th solutions in the responses of all instances in the test subset.[1] The distinctness ratio is defined as:

$$\mathrm{Dis}^m = \frac{\sum_{k=1}^{K} \tau_k^m}{K}$$

where

$$\tau_k^m = \begin{cases} 1, & \text{if } \Phi(s_k^m) \nsubseteq \{\Phi(s_k^1), \ldots, \Phi(s_k^{m-1})\} \\ 0, & \text{otherwise} \end{cases}$$

In this context, $\Phi(s_k^m)$ is the reasoning strategy of $s_k^m$. Intuitively, $\tau_k^m = 1$ if the $m$-th solution introduces a new reasoning strategy, and $\mathrm{Dis}^m$ is the average number of new reasoning strategies at solution position $m$. We use GPT-4o to cluster the solutions for each instance into groups via a prompt like (Ye et al., 2024).[2]

Figure 7 displays the distinctness ratio for each solution index. Intuitively, the ratio for Solution#1 is always 100%, as it has no preceding solutions, thus $\tau \equiv 1$ for all instances. Generally, the ratio decreases with higher indices, indicating that **later solutions often repeat earlier ones**. For example, the average distinctness ratio for Solution#≥4 across test sets decreases by 7.5% compared to Solution#3. The ratio of Solution#2 significantly decreases compared to Solution#1. By reviewing outputs, we find that Solution#2 often double-checks answers from Solution#1 using the same reasoning strategy. Subsequently, Solution#3 attempts to solve the problem using a new reasoning strategy.

---

[1]If a response does not contain the $m$-th solution (i.e. $M_i < m$), that response is excluded from the set, hence $K$ does not necessarily equal the number of test set instances $N$.

[2]Refer to Appendix A.1 for clustering prompt and results.

**Process Efficiency Metric** Based on the above observation, we propose a process efficiency metric to empirically evaluate the contribution of later solutions to solution diversity. The process efficiency metric is calculated as:

$$\xi_P = \frac{1}{N} \sum_{i=1}^{N} \frac{D_i}{T_i} \tag{2}$$

where $D_i$ represents the number of efficient tokens that contribute to the solutions' diversity. Here, we intentionally exclude the factor $\sigma_i$ to concentrate on diversity, **independent of correctness**. Let $T_i^m$ denote the number of tokens in solution $s_i^m$. We define:

$$D_i = \sum_{m=1}^{M} \tau_i^m T_i^m$$

Intuitively, the tokens in a distinct solution are regarded as process efficient tokens, and $\xi_P$ is the average ratio of tokens associated with diverse reasoning strategies. In the example shown in Figure 2, the 13 solutions are categorized into 7 distinct reasoning strategies. Consequently, tokens in Solutions 1, 2, 3, 7, 8, 9, and 12 are efficient, resulting in $\xi_P = \frac{(39+109+39+29+29+19+59)}{901} = 35.8\%$.

### 2.4. Empirical Efficiency Results

Table 1 presents the results on model efficiency. For comparison, we include two representative conventional LLMs: Llama-3.3-70B-Instruct and Qwen2.5-Math-72B-Instruct. These conventional LLMs produce only a single solution, meaning that $\frac{D_i}{T_i} = \frac{\hat{T}_i}{T_i} = 1$. Therefore, in these cases, the outcome efficiency metric $\xi_O = \frac{1}{N} \sum_{i=1}^{N} \sigma_i$ equals accuracy, and the process efficiency metric $\xi_P = 1.0$. In comparison, long reasoning models generate significantly longer responses, which are less efficient in improving accuracy and solution diversity. We refer to the inefficient use of generated tokens as the "**overthinking issue**". The

*Table 1.* Model efficiency results of strong LLMs.

| Models | Accuracy | Response | | Efficiency | |
|---|---|---|---|---|---|
| | | #Solution | #Token | Outcome | Process |
| *ASDIV* | | | | | |
| Llama-3.3-70B-Instruct | 95.6 | 1.0 | 166.4 | 95.6% | 100.0% |
| Qwen2.5-Math-72B-Instruct | 96.3 | 1.0 | 213.0 | 96.3% | 100.0% |
| Qwen/QwQ-32B-Preview | 96.5 | 3.5 | 713.7 | 53.7% | 69.0% |
| Qwen/QwQ-32B | 97.1 | 3.4 | 1171.1 | 57.6% | 69.1% |
| Qwen/Qwen3-235B-A22B | 97.2 | 3.8 | 1399.2 | 56.3% | 67.9% |
| DeepSeek/R1 | 97.1 | 3.3 | 840.0 | 57.5% | 70.2% |
| *GSM8K* | | | | | |
| Llama-3.3-70B-Instruct | 92.6 | 1.0 | 220.3 | 92.6% | 100.0% |
| Qwen2.5-Math-72B-Instruct | 95.8 | 1.0 | 317.4 | 95.8% | 100.0% |
| Qwen/QwQ-32B-Preview | 94.7 | 2.9 | 756.1 | 11.8% | 75.2% |
| Qwen/QwQ-32B | 96.1 | 4.2 | 1835.8 | 53.3% | 61.8% |
| Qwen/Qwen3-235B-A22B | 95.7 | 5.3 | 2419.4 | 47.9% | 58.6% |
| DeepSeek/R1 | 96.4 | 2.9 | 1053.1 | 67.2% | 73.2% |
| *MATH500* | | | | | |
| Llama-3.3-70B-Instruct | 75.4 | 1.0 | 553.4 | 75.4% | 100.0% |
| Qwen2.5-Math-72B-Instruct | 86.8 | 1.0 | 593.1 | 86.8% | 100.0% |
| Qwen/QwQ-32B-Preview | 91.2 | 4.3 | 2398.5 | 51.4% | 70.3% |
| Qwen/QwQ-32B | 96.2 | 6.2 | 4365.4 | 44.4% | 61.1% |
| Qwen/Qwen3-235B-A22B | 92.6 | 6.2 | 5810.2 | 43.5% | 60.9% |
| DeepSeek/R1 | 95.4 | 4.2 | 2696.4 | 53.0% | 70.7% |

experimental results demonstrate that while long reasoning models have the capacity to generate multiple solutions, their efficiency is hindered by the overthinking issue.

Figure 8 presents the detailed efficiency results of DeepSeek-R1 across various difficulty levels of the MATH500 test set. Notably, DeepSeek-R1 performs poorly on the simplest Level 1 problems, achieving about 50% outcome efficiency. These findings underscore that **the overthinking issues faced by long reasoning models are pronounced with simpler math problems**.

## 3. Mitigating Overthinking Issues

In this section, we explore strategies to improve the efficiency of long reasoning models. Our approach involves three key steps: (1) generating long reasoning traces on a mathematical dataset, (2) constructing shorter reasoning traces by pruning unnecessary solutions from the long traces, and (3) applying preference optimization to train the model on (short, long) reasoning pairs. Following prior work, we use the PRM12K dataset (Lightman et al., 2024) as our training set. Our experiments are conducted on the QwQ-32B-Preview model, leveraging a self-training strategy (Zelikman et al., 2022; Ho et al., 2023) in which the model generates its own training data.

*Table 2.* Statistics on different types of generated responses based on the training data. "Greedy" denotes responses generated via greedy search; "Shortest" and "Longest" indicate the shortest and longest responses among 10 samples, respectively.

| Response | #Solution | #Token | Efficiency | |
|---|---|---|---|---|
| | | | Outcome | Process |
| Greedy | 3.1 | 1434.8 | 55.6% | 72.6% |
| Shortest | 2.5 | 1051.3 | 69.8% | 80.3% |
| Longest | 4.1 | 2258.7 | 46.0% | 66.4% |

### 3.1. Generating Long Reasoning Traces

We begin by generating long reasoning traces for the PRM12K dataset through sampling. Using the QwQ-32B-Preview model with a temperature of 1.0, we produce 10 reasoning traces per question and retain only those yielding correct answers. Table 2 presents the statistics of different types of generated responses. Our analysis of these sampled responses reveals that the shortest response performs better in terms of both outcome and process efficiency, using fewer rounds and tokens. These findings support our initiative to enhance model efficiency through simplifying responses.

*Table 3.* Statistics on different types of positive examples. "#S" denotes the number of solutions.

| Positive Example | #S | #Token | Efficiency | |
|---|---|---|---|---|
| | | | Outcome | Process |
| Shortest Response | 2.5 | 1051.3 | 69.8% | 80.3% |
| FCS | 1.1 | 681.0 | 99.5% | 99.1% |
| FCS + Ref. | 1.9 | 878.7 | 78.4% | 82.4% |
| GDS | 1.6 | 856.8 | 86.8% | 94.2% |

### 3.2. Simplifying Responses to Enhance Efficiency

Building on the observations in Section 2, where earlier solutions demonstrate greater efficiency, we propose three strategies for simplifying response generation by constructing concise reasoning traces:

- **First-Correct Solutions (FCS)**: We retain the portion of the response from the beginning until the first occurrence of a correct answer.

- **FCS+Reflection**: Since most responses yield the correct answer in the first solution (see Figure 6), retaining only the First-Correct Solutions might cause long-reasoning models to revert to conventional LLM behavior. To mitigate this, we extend the approach by including the second correct solution, thereby preserving the model's reflective capability while maintaining efficiency.

- **Greedily Diverse Solutions (GDS)**: Figure 7 shows that the distinctiveness of the second solution diminishes significantly, as it often double-checks the first solution using the same reasoning strategy. Consequently, FCS+Reflection may compromise process efficiency. To address this, we introduce a simple heuristic that greedily selects solutions with new reasoning strategies. This approach would include additional solutions when the second solution introduces novel reasoning, thereby enhancing diversity.

We also use the **Shortest Response** among the sampling results as the baseline of simplifying responses.

Table 3 presents the statistics for these simplified responses. Notably, all simplified responses enhance efficiency compared to the shortest response. "FCS" is the most efficient, both in terms of outcome and process, using the fewest number of solution rounds and tokens. "FCS+Reflection" incorporates reflection, requiring approximately one additional solution round, which reduces both outcome and process efficiencies. "Greedily Diverse Solutions" serves as a compromise, balancing the number of solutions and tokens, and achieving moderate to high efficiency.

### 3.3. Length Preference Optimization

For each question, we construct a paired training dataset where:

- The positive example is the short reasoning trace

- The negative example is the longest response from the sampled outputs

We then apply preference optimization to these pairs, training the model to favor concise, efficient reasoning paths. Our experiments focus on the following approaches:

- **Direct Preference Optimization** (DPO; Rafailov et al. 2024): This method trains a model directly on human-preferred responses to increase the likelihood of preferred responses over unpreferred ones.

- **Reasoning Preference Optimization** (RPO; Pang et al. 2024; Liu et al. 2024): This approach modifies the DPO loss by adding a negative log-likelihood term on the preferred response. RPO enhances DPO training stability by preventing a decreased probability of selected responses.

- **Simple Preference Optimization** (SimPO; Meng et al. 2024): This method addresses the discrepancy between the reward function and the generation metric during inference found in other preference optimization methods. SimPO incorporates techniques like adaptive margin and length regularization into DPO training.

We also explore **Supervised Fine-Tuning** (SFT; Wei et al. 2022a) on the positive examples, which enables the model to directly learn how to generate shorter responses.

## 4. Experiments

### 4.1. Datasets

We use the following test sets for our experiments:

- **ASDIV** (Miao et al., 2020): an English math word problem corpus with 2,305 instances, typically found in **elementary schools**.

- **GSM8K** (Cobbe et al., 2021): a dataset of high-quality, linguistically diverse **grade school math word problems** created by human problem writers. The test set includes 1,319 problems, with solutions often involving a sequence of elementary calculations using basic arithmetic.

- **MATH500** (Hendrycks et al., 2021): a challenging dataset consisting of problems from **high school math competitions** across seven subjects (e.g., Prealgebra, Algebra, Number Theory) and difficulty levels based on AoPS (ranging from 1 to 5).

*Table 4.* Experimental results of the proposed efficiency enhancing methods.

| Methods | Accuracy | Response | | Efficiency | |
|---|---|---|---|---|---|
| | | #Solution | #Token | Outcome | Process |
| *ASDIV* | | | | | |
| QwQ-32B-Preview | 96.5 | 3.5 | 713.7 | 53.7% | 69.0% |
| +SimPOFCS+Reflection | 96.6 | 1.9 | 381.5 | 82.5% | 87.9% |
| *GSM8K* | | | | | |
| QwQ-32B-Preview | 94.7 | 2.9 | 756.1 | 11.8% | 75.2% |
| +SimPOFCS+Reflection | 95.9 | 1.8 | 416.5 | 86.0% | 91.0% |
| *MATH500* | | | | | |
| QwQ-32B-Preview | 91.2 | 4.3 | 2398.5 | 51.4% | 70.3% |
| +SFTShortest Response | 92.6 | 4.4 | 2359.0 | 59.7% | 72.8% |
| +DPOShortest Response | 93.2 | 3.4 | 1928.8 | 64.3% | 77.8% |
| +RPOShortest Response | 90.2 | 3.5 | 2015.2 | 64.7% | 76.6% |
| +SimPOShortest Response | 91.0 | 3.5 | 1871.5 | 64.7% | 78.1% |
| +SimPOFirst-Correct Solution | 90.4 | 1.3 | 1015.6 | 85.5% | 96.3% |
| +SimPOFCS+Reflection (Ours) | 91.4 | 2.4 | 1330.3 | 79.1% | 88.9% |
| +SimPOGreedily Diverse Solutions | 91.2 | 1.7 | 1285.8 | 80.1% | 90.2% |
| *GPQA* | | | | | |
| Qwen2.5-Math-72B-Instruct | 46.5 | 1.0 | 811.7 | 46.5% | 100% |
| QwQ-32B-Preview | 59.1 | 6.0 | 3226.4 | 38.6% | 66.4% |
| +SimPOFCS+Reflection | 59.6 | 3.0 | 2084.5 | 57.9% | 80.2% |
| *AIME24* | | | | | |
| Qwen2.5-Math-72B-Instruct | 23.3 | 1.0 | 1204.5 | 23.3% | 100.0% |
| QwQ-32B-Preview | 46.7 | 3.7 | 9480.6 | 37.5% | 79.5% |
| +SimPOFCS+Reflection | 43.3 | 2.0 | 5154.4 | 39.3% | 91.6% |

The overall difficulty levels of the test sets are ASDIV < GSM8K < MATH500.

We also validate our method using more challenging test sets, specifically GPQA and AIME:

- **GPQA** (Rein et al., 2023): a **graduate-level** dataset consisting of multiple-choice questions in subdomains of physics, chemistry, and biology. For our experiment, we select the highest quality subset, known as GPQA Diamond (composed of 198 questions).

- **AIME24** (MAA Committees): a dataset from the American Invitational **Mathematics Examination**, which tests mathematical problem solving across multiple areas (e.g. algebra, counting, geometry, number theory, and probability).

### 4.2. Experimental Results

Table 4 presents the main results of the proposed methods. We first conduct a detailed comparison on MATH500 to identify the most effective approach, and then validate the proposed method on additional benchmarks.

**Performance of Length Preference Optimization Methods** We first evaluate all methods described in Section 3.3 under the Shortest Response setting. While supervised fine-tuning (SFT) yields modest reductions in the number of solutions and tokens compared to the vanilla QwQ-32B-Preview model, preference optimization methods demonstrate significantly greater effectiveness. Among these, SimPO achieves the best performance, reducing generated tokens by 22% on MATH500 while maintaining comparable solution quality. Based on these results, we select SimPO as our default method for subsequent experiments.

**Performance of Response Simplification Methods** As anticipated, the First-Correction Solutions (FCS) strategy achieves the greatest length reduction. However, this approach degrades performance on the challenging MATH500 test set, likely due to its need for more extensive reflection. The FCS+Reflection variant addresses this limitation, outperforming standard FCS through an additional reflection round. While the Greedily Diverse Solutions strategy maintains a balance between performance and token efficiency, it underperforms FCS+Reflection - supporting our hypothesis that the difficult MATH500 problems ben-

efit from the deep inference capabilities of long reasoning models. We consequently establish FCS+Reflection as our default response simplification method. The performance of the FCS+Reflection method across different test sets demonstrates its effectiveness. Accuracy improves slightly on ASDIV, GSM8K, and MATH500, with nearly half the tokens and solutions compared to the QwQ-32B-Preview baseline. Additionally, FCS+Reflection enhances efficiency, as reflected in the efficiency metrics.

**Results on Challenging Test Sets**  Our approach significantly improves efficiency on simpler benchmarks (ASDIV and GSM8K) while maintaining comparable performance. However, a critical concern remains: could this length preference optimization method compromise the models' capacity for complex, long-term reasoning? To directly address this, we conduct evaluations on challenging datasets requiring extended reasoning chains (GPQA and AIME24). The results demonstrate that our approach preserves the models' complex reasoning capabilities while reducing token usage - performance on GPQA and AIME24 only drops slightly with improved efficiency. This confirms that our method achieves robust generalizability without sacrificing the fundamental strengths of long reasoning models.

## 5. Related Work

### 5.1. Scaling Test-Time Compute

Enhancing model performance on complex tasks can be achieved by scaling test-time compute, which involves:

**Expanding Search Space**  LLMs have strong reasoning abilities, but their auto-regressive decoding often misses optimal solutions. Self-consistency generates multiple responses and use majority voting to select the best answer (Wang et al., 2023b). Other approaches include best-of-n decoding, minimum Bayes risk decoding (Lightman et al., 2024; Li et al., 2023; Khanov et al., 2024; Heineman et al., 2024; Wu et al., 2024), and structured search methods such as Tree-of-Thought, Graph-of-Thought, and Monte Carlo Tree Search (Yao et al., 2024; Besta et al., 2024; Luo et al., 2024; Tian et al., 2024; Wan et al., 2024).

**Human-Like Thinking Patterns**  LLMs often use natural language reasoning. Techniques like chain-of-thought encourage step-by-step reasoning instead of direct answers (Wei et al., 2022b; Kojima et al., 2022). This has been expanded with methods like debating, self-correction, self-critique, and plan-and-solve (Liang et al., 2024; Du et al., 2024; Xiong et al., 2023; Kumar et al., 2024; Kamoi et al., 2024; Ke et al., 2023; Lin et al., 2024; Yu et al., 2024; Wang et al., 2023a). Recent studies also explore latent space reasoning to mimic human cognition (Hao et al., 2024; Goyal

et al., 2024). Advanced models combine these patterns into extensive chains-of-thought, improving accuracy with more reasoning time (OpenAI, 2024).

### 5.2. Efficient Thinking

Scaling the search space and scaling human-like thinking involves two distinct aspects of efficiency: efficient search and efficient thinking. However, few works specifically focus on efficient thinking in LLMs. Kimi et al. (2025) leveraged the long to short strategy to compress generation context. Zhao et al. (2024) encourages the model to terminate reasoning by saying "I don't know" when the problem is hard to solve. Han et al. (2024) introduces token-budget-aware reasoning, where the model is prompted with a specified token budget to guide its reasoning process. There are also several contributions (Damani et al., 2024; Wang et al., 2024; Xu et al., 2024) made to predict the distribution of the computation budget and allocate the computation power based on the prompt's difficulty. Another line of work emphasizes the early stopping strategy to save computation budget while reasoning (Manvi et al., 2024; Li et al., 2024). Moreover, multi-agent framework utilizes large LLMs for difficult tasks while small LLMs for simple tasks (Kirchner et al., 2024; Damani et al., 2024)

In summary, all the aforementioned works consider conventional models rather than long reasoning models with longer chains-of-thought. In contrast, our work first identifies the overthinking problem in long reasoning model. Additionally, instead of limiting the reasoning space or leaving the token budget to be specified by the user, we aim to train the model to learn how to think efficiently.

## 6. Conclusion

This study identifies a key challenge in long reasoning models —- efficient and intelligent scaling of test-time computational resources. We have presented a comprehensive analysis of the overthinking issue in long reasoning models. By highlighting the overthinking phenomenon and proposing efficiency metrics, we enhance our understanding of resource utilization in o1-like models. Our self-training based approach effectively mitigates overthinking, reducing unnecessary computation while maintaining performance across reasoning benchmarks of varying difficulty levels.

This work not only boosts efficiency in long-reasoning models, but also lays critical groundwork for future research aimed at optimal allocation of computational resources in AI reasoning tasks, which is crucial for the development of sustainable and cost-effective AI systems. Future directions include exploring adaptive compute strategies that dynamically adjust to problem complexity and refining efficiency metrics for broader model generalization.

## Impact Statement

The paper reveals the overthinking issue in long reasoning models. Our findings call for the community to research on models' efficient thinking capabilities, which could significantly influence future developments in this field. We see no harmful impacts of this work.

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

# A. Appendix

## A.1. Prompts for Clustering Solutions

Inspired by (Ye et al., 2024), we leverage GPT-4o to cluster the solutions for each instance into groups with the following prompt:

---

Criteria for clustering the mathematical solutions:

1. If the solutions used to arrive at the solutions are fundamentally different from each other, such as algebraic manipulation versus geometric reasoning, they can be considered novel;

2. Even if the results are the same, if the intermediate steps or processes involved in reaching those solutions vary significantly, the solutions can be considered different;

3. If the solutions relies on different assumptions or conditions, they should be considered different from each other;

4. A solution might generalize to a broader class of problems, while another solution might be specific to certain conditions. In such cases, they are considered distinct;

5. If one solution is significantly simpler or more complex than the others, it can be regarded as essentially novel, even if they lead to the same result.

Given the following mathematical problem:
***problem***

Solutions:
Solution 1: ...
Solution 2: ...

Please output the clusters strictly following the following format, each row containing a cluster, names, and reasons. Do not include any additional text or explanations outside of this format:
cluster1 [solution names] reason for cluster
cluster2 [solution names] reason for cluster
cluster3 [solution names] reason for cluster
...

For example:
cluster1 [Solution 1, Solution 3, Solution 5] similar algebraic approach using the volume formula and canceling terms or directly solving for the height.
cluster2 [Solution 2, Solution 4] verifying the correctness and consistency of the formula and solution and considering unit checks or logical reasoning on how volume relates to base area and height.

---

The clustering results for the QwQ-32B-Preview response in Figure 2 are:

---

cluster1 [Solution 1, Solution 6, Solution 11] stating or affirming the basic arithmetic fact that 2 plus 3 equals 5.

cluster2 [Solution 2, Solution5] breaking down the addition into smaller, simpler steps to reach the result.

cluster3 [Solution 3, Solution 4] using a practical analogy of counting objects to explain the addition.

cluster4 [Solution 7] using subtraction as a reverse check to verify the addition result.

cluster5 [Solution 8] using algebraic manipulation and solving simple equations to confirm the result.

cluster6 [Solution 9, Solution 10] converting numbers into different systems (binary and Roman numerals) to verify the result.

cluster7 [Solution 12, Solution 13] considering specific contexts or frameworks like modular arithmetic or programming which could change traditional addition results.

---

