# OpenReview forum: "Do NOT Think That Much for 2+3=? On the Overthinking of Long Reasoning Models"
_ICML.cc/2025/Conference — ICML 2025 poster_

### Official Review · Reviewer_yXZs · 2025-03-06

**Overall Recommendation:** 3

**Summary:**

The authors observe that these models often allocate excessive computational resources to simple problems, leading to inefficiencies. To address this, they introduce novel efficiency metrics from both outcome and process perspectives and propose self-training strategies to streamline reasoning without compromising accuracy. Experimental results demonstrate that their approach reduces computational overhead while preserving model performance across various test sets.

## update after rebuttal

I will keep my positive rating.

**Claims And Evidence:**

The primary claim is that o1-like LLMs exhibit overthinking, especially on simple problems, resulting in unnecessary computational resource usage. The authors support this by analyzing model responses to simple arithmetic questions, where o1-like models consumed significantly more tokens than conventional models to reach the same answer. They also introduce efficiency metrics to quantify this overthinking and demonstrate that their self-training strategies can mitigate it without sacrificing accuracy.

**Essential References Not Discussed:**

NA

**Experimental Designs Or Analyses:**

Yes. The experimental design involves evaluating o1-like models on various mathematical benchmarks to assess overthinking patterns. The authors introduce efficiency metrics to quantify overthinking and apply self-training strategies to mitigate it. They demonstrate that their approach reduces computational overhead while preserving model performance across different test sets. In general, the experiments are solid and convincing.

**Methods And Evaluation Criteria:**

The authors employ a self-training paradigm to reduce overthinking, streamlining reasoning processes without compromising accuracy. They evaluate their approach using novel efficiency metrics from both outcome and process perspectives, assessing the rational use of computational resources by o1-like models. Experimental results show that their approach successfully reduces computational overhead while preserving model performance across a range of test sets with varying difficulty levels.

**Other Comments Or Suggestions:**

NA

**Other Strengths And Weaknesses:**

A potential weakness is the generalization of the proposed phenomenon and algorithm. More models and general evaluations can be helpful.

**Questions For Authors:**

Do you have the results for the proposed efficiency enhancing methods with more models?

Have you checked the evaluation on other benchmarks beyond mathematical/scientific reasoning?

**Relation To Broader Scientific Literature:**

This work contributes to the broader scientific literature by addressing the inefficiencies in o1-like LLMs due to overthinking. By introducing efficiency metrics and self-training strategies, the authors provide insights into optimizing computational resource allocation during inference, aligning with ongoing research on improving the efficiency and effectiveness of LLMs.

**Theoretical Claims:**

The paper does not focus on theoretical claims but rather on empirical observations and practical solutions to the overthinking problem in o1-like LLMs.

---

> ### Author Rebuttal · Authors · 2025-04-01
>
> Thank you for your questions! We will answer the questions mentioned in your review.
>
> Q: Do you have the results for the proposed efficiency enhancing methods with more models?
>
> A: Yes, we have conducted experiments on the R1-Distilled-Qwen-32B model, and our efficiency-enhancing methods remain effective.
>
> | Model                 | Dataset | Acc  | #Solutions | Length | Outcome Efficiency | Process Efficiency |
> | --------------------- | ------- | ---- | ---------- | ------ | ------------------ | ------------------ |
> | R1-Distilled-Qwen-32B | ASDIV   | 96.2 | 1.6        | 308.6  | 84.4               | 95.0               |
> | +SimPO+FCS+Reflection | ASDIV   | 96.4 | 1.2        | 209.1  | 87.0               | 97.4               |
> |                       |         |      |            |        |                    |                    |
> | R1-Distilled-Qwen-32B | GSM8k   | 94.1 | 1.3        | 328.3  | 80.9               | 96.4               |
> | +SimPO+FCS+Reflection | GSM8k   | 93.3 | 1.1        | 254.4  | 82.5               | 98.1               |
> |                       |         |      |            |        |                    |                    |
> | R1-Distilled-Qwen-32B | MATH500 | 90.2 | 4.7        | 3210.0 | 54.2               | 71.0               |
> | +SimPO+FCS+Reflection | MATH500 | 91.4 | 3.4        | 2735.2 | 63.9               | 77.2               |
>
>
> ---
> ---
> Q: Have you checked the evaluation on other benchmarks beyond mathematical/scientific reasoning?
>
> A: We have not performed evaluations on benchmarks beyond mathematical and scientific reasoning. There are two primary reasons behind this choice: (1) Our paper specifically focuses on mathematical and scientific reasoning tasks, as they represent cutting-edge domains that strongly reflect aspects of human intelligence. (2) Benchmarks in general domains often lack clearly defined and verifiable answers, which makes objective evaluation challenging.
>
> Investigating the overthinking phenomenon in more general tasks by leveraging an LLM-as-a-judge evaluation approach is an interesting direction that we intend to explore in future work.

---

### Official Review · Reviewer_JarS · 2025-03-10

**Overall Recommendation:** 4

**Summary:**

This paper addresses the "overthinking" issue observed in o1-like reasoning models, which expend excessive computational resources during inference, especially for simple problems. The authors first analyze this phenomenon by prompting LLMs with trivial questions (e.g., "What is the answer of 2 plus 3?"), observing significant redundancy in generated solutions. They identify that models often produce numerous redundant or minimally diverse solutions, which do not substantially contribute to improving accuracy. To quantify this, the authors propose novel efficiency metrics from both outcome (accuracy contribution) and process (solution diversity) perspectives. Leveraging these metrics, the authors introduce a self-distillation approach that samples diverse solutions via temperature sampling and recombines selected efficient solutions to streamline inference. Experiments across benchmarks (e.g., GSM8K, MATH500, GPQA, AIME) demonstrate the proposed approach significantly reduces token usage while maintaining accuracy, highlighting its effectiveness in enhancing CoT efficiency.

**Claims And Evidence:**

The claims presented in the paper are generally supported by clear and convincing evidence. Specifically, the paper makes three primary claims:
(1) o1-like large language models exhibit a significant "overthinking" issue, especially on simple problems.
(2) novel efficiency metrics (outcome and process metrics) effectively quantify this inefficiency.
(3) self-distillation strategies leveraging these metrics substantially mitigate overthinking without compromising performance.

**Essential References Not Discussed:**

None.

**Experimental Designs Or Analyses:**

The experimental designs and analyses presented in the paper are generally sound and valid. The authors provide thorough and rigorous experiments across multiple well-established benchmark datasets of varying difficulty (ASDIV, GSM8K, MATH500, GPQA, and AIME), clearly demonstrating the effectiveness of their proposed methods in mitigating overthinking.

**Methods And Evaluation Criteria:**

The proposed methods and evaluation criteria effectively address the stated problem. The authors introduce intuitive outcome and process efficiency metrics, clearly capturing the extent and nature of the "overthinking" phenomenon in o1-like models. Their chosen benchmark datasets (ASDIV, GSM8K, MATH500, GPQA, AIME) are appropriate and well-aligned with the goal of assessing reasoning efficiency across a range of difficulty levels. Additionally, the self-training approaches (SimPO combined with streamlined solutions like FCS+Reflection) logically target inefficiencies identified by the metrics, making them well-suited for the study's objectives. Overall, both the methods and evaluation criteria are well-motivated and sensible for the problem at hand.

**Other Comments Or Suggestions:**

None.

**Other Strengths And Weaknesses:**

Strengths:

1.The paper addresses a novel and practically significant problem—the "overthinking" inefficiency in o1-like LLMs—providing valuable insights into the intelligent allocation of inference-time resources.

2.The proposed metrics (outcome and process efficiency) are original, clearly defined, and intuitively appealing.

3.Experiments are comprehensive, covering multiple datasets and difficulty levels, enhancing confidence in the robustness of the findings.


Weaknesses:

1.While the metrics and simplification methods are effective, the generalizability to domains beyond mathematical reasoning tasks remains unclear.

**Questions For Authors:**

The paper does not discuss the "self-reflection" or "aha moment" behavior recently appeared in "Guo, Daya, et al. "Deepseek-r1: Incentivizing reasoning capability in llms via reinforcement learning." (2025).", which describes LLMs revising earlier reasoning after reflecting or backtracking.

Question 1:
Could the authors clarify how they categorize this type of reflective or "aha moment" behavior in their framework?

Question2:
Should a solution that emerges from a reflective or backtracking process be retained during the self-distillation process, or would it be considered redundant and thus removed?

**Relation To Broader Scientific Literature:**

The paper clearly situates its contributions within the broader literature. While prior research has extensively studied methods like self-consistency, Tree-of-Thought, and early-stopping to optimize test-time computation and improve model reasoning, this paper uniquely identifies the "overthinking" issue specific to o1-like LLMs. The authors bridge a relevant gap by proposing targeted efficiency metrics and self-training strategies (SimPO, FCS+Reflection) that explicitly address inefficiencies in extended chains-of-thought.

**Theoretical Claims:**

This is not a theory paper. It conducts emperical study.

---

> ### Author Rebuttal · Authors · 2025-04-01
>
> Thank you for your insightful questions. We address each question raised in your review individually below.
>
> Q (Weakness): While the metrics and simplification methods are effective, the generalizability to domains beyond mathematical reasoning tasks remains unclear.
>
> A: This is a valuable point. In this paper, we primarily focus on the mathematical reasoning domain because verifying the correctness of answers to mathematical questions is straightforward and objective. Nevertheless, we have noticed similar "overthinking" phenomena in general reasoning tasks, such as repeatedly asking the same questions or recurring assumptions. A deeper and more comprehensive analysis of such phenomena in more general tasks is warranted, which we plan to explore in future work.
>
> ---
> Q: Could the authors clarify how they categorize this type of reflective or "aha moment" behavior in their framework?
>
> A: The "aha moment" described in R1-Zero refers to a training-time behavior, in which a model autonomously acquires reasoning strategies via reinforcement learning. In contrast, our framework specifically targets inference-time behaviors in models executing long reasoning processes. In our work, a "solution" explicitly denotes portions of the model-generated outputs containing a final answer. Therefore, the concept of the "aha moment" aligns more closely with reflective behaviors (e.g., reflection, self-verification, or backtracking) that bridge two solutions. When the model continues reasoning beyond obtaining an answer within a solution — whether through reflection, self-verification, or backtracking — we characterize this behavior as "overthinking".
>
> ---
> Q: Should a solution that emerges from a reflective or backtracking process be retained during the self-distillation process, or would it be considered redundant and thus removed?
>
> A: Whether a reflective or backtracking-derived solution should be retained during self-distillation depends on the specific training data construction policy described in Section 3.2. For example, our "First Correct Solution" (FCS) policy retains only the first solution generated, whereas the "FCS+Reflection" policy retains one additional reflective solution alongside the initial FCS solution.

---

> > ### Comment · Reviewer_JarS · 2025-04-03
> >
> > Thank you for the clarification :D ! I enjoy reading your paper!
> >
> > Q1:
> >
> > I agree.
> >
> > Q2:
> >
> > Not fully agree.
> >
> > a,I do not think the "aha moment" only appear in the training-time.
> >
> > b,The "aha moment" happens when a model backtracking from an incorrect answer to a correct answer.
> >
> > Therefore, should "aha moment" be treated as "overthinking"?
> >
> > Q3:
> >
> > Your answer is self-consistent. (no problem!)

---

> > > ### Author Response · Authors · 2025-04-04
> > >
> > > Thank you for your constructive feedback. Below, we address Q2 with additional detail and clarification.
> > >
> > > To align with standard terminology, we will adopt the term ​"reasoning strategies" (e.g., reflection, self-verification, backtracking) in place of the "aha moment" in this discussion. These reasoning strategies do exist in both ​training-time and ​inference-time.
> > >
> > > We define "overthinking" as a phenomenon where a long reasoning model repeatedly evaluates and revisits solutions within the CoT reasoning process, rather than converging promptly to a final answer. Crucially, this differs from scenarios where the model backtracks from an incorrect answer to a correct one—a valid corrective process. Overthinking specifically arises when the model redundantly applies strategies like reflection, self-verification, or backtracking ​after already arriving at the same answer in successive iterations.
> > >
> > > We appreciate your engagement with this topic. Your feedback has deepened our conceptual framing of overthinking and will help refine its articulation in subsequent work.

---

### Official Review · Reviewer_xCey · 2025-03-12

**Overall Recommendation:** 3

**Summary:**

This paper studies the over-thinking problem of o1-like models. The overthinking refers to the scenario where unnecessary compute resource is used for simple question.
It first compares the average tokens needed by traditional LLM and the o1-like reasoning models. It finds that o1-like models spend much more tokens on simple problems with minimal benefits and lack diversity in reasoning strategies. Based on this observation, it proposes several metrics to evaluate the efficiency of LLM reasoning models
It then adopts a self-training paradigm to mitigate this issue by removing redundant solutions. Results on GSM8K, MATH500, GPQA, and AIME demonstrate effectiveness and robustness

**Claims And Evidence:**

This paper claims that o1-like reasoning models have the overthinking issue. It investigates this issue based on the proposed Outcome Efficiency Metric, distinctness ratio, and Process Efficiency Metric. Their claims are supported by the comparisons between traditional LLM and o1-models, which shows that their longer responses are less efficient in improving accuracy and solution diversity.

**Essential References Not Discussed:**

Necessary references have been discussed in their related work

**Experimental Designs Or Analyses:**

The experiments in Section 3 lack the necessary details
	• What are the sizes of different kinds of training examples in Table 2 and Table 3?
	• What are the training details (such as hyperparameters) for models in Section 3?

The descriptions in Section 3.1 and 3.2 are confusing. Which part is the method proposed by this paper?
	• Section 3.1 lists several self-training methods such as SFT and DPO, but these methods are not particularly designed for Length Preference Optimization. What does Length Preference Optimization refer to here? Is there a specific dataset for length optimization?
	• The purpose of Streamlining Responses in Section 3.2 is unclear. Are they used during the inference time to select the best response? Or are they the strategies for constructing self-training datasets?

Why experimental settings in different datasets are different? For example, on MATH500, there are 7 settings, while for other datasets, there is only one. Similarly, Qwen2.5-Math-72B-Instruct is one of the baselines for GPQA and AIME24, but their performance on ASDIV, GSM8k, MATH500 is not mentioned.

What does "ours" mean in SimPO FCS+Reflection (Ours) ? It looks like this one is the final method proposed by this paper and the other settings are just baseline or ablation studies. If that is the case, this should be explicitly mentioned.

**Methods And Evaluation Criteria:**

To investigate the overthinking, it proposes new evaluation metrics Outcome Efficiency Metric, distinctness ratio, and Process Efficiency Metric to evaluate the efficiency and diversity of the solution. One concern for the diversity is that it is independent of the correctness. High diversity but an incorrect final response may not make too much sense.

To mitigate overthinking, it finetunes the model towards short and correct solutions. It also finetunes the model on simplified responses.

**Other Comments Or Suggestions:**

Section 3 should be reorganized to make it clearer. Necessary experimental details should be added for reproduction.

**Other Strengths And Weaknesses:**

Strength: The detailed and comprehensive investigations on the overthinking scenario of o1-like models provide more insight into the performance of o1-like models.

Weakness: The methods and experiments to mitigate the overthinking issue are not convincing.

**Questions For Authors:**

In the Outcome Efficiency Metric and Process Efficiency Metric, why does the first consider correctness and the second one independent of correctness?

**Relation To Broader Scientific Literature:**

The overthinking issue is interesting and important for analyzing the performance of current o1-like models. It provides insights on how many of the tokens in the extremely long chain of thought contribute to the final accuracy.

**Theoretical Claims:**

There is no theoretical claim in the paper.

---

> ### Author Rebuttal · Authors · 2025-04-01
>
> Thank you very much for your questions. Below we address each point mentioned in your review.
>
> Since most of your questions pertain to Section 3, we provide additional clarifications on our methods as follows:
>
> 1. Our proposed methods focus primarily on constructing suitable training data. Specifically, Section 3.2 describes how we generate paired training data for preference optimization algorithms.
>
> 2. Section 3.1 describes the preference optimization algorithms employed in our experiments. As you pointed out, these algorithms were not originally designed for length optimization. Nevertheless, we explore their potential effectiveness in addressing the length optimization task.
>
> 3. We evaluated the performance of various preference optimization algorithms across different training datasets. Ultimately, we chose SimPO trained with the FCS+Reflection dataset as our final method, marked as "ours" in Table 4.
>
> 4. The total number of training examples is approximately 11,000. Both SFT and preference optimization methods were trained using a learning rate of 1e-7, with a cosine learning rate scheduler, and a batch size of 16. We trained the SFT models for four epochs and the preference optimization models for one epoch. We will include these details in the appendix of the next version.
>
> 5. For simplicity and clarity, we first compared multiple training algorithms using the "Shortest Response" dataset, finding SimPO performed the best. Subsequently, we tested SimPO on a broader range of training datasets and selected "FCS+Reflection" as our best performing dataset configuration. The complete experimental results will be provided in the appendix in our revised manuscript.
>
> Thank you again for your insightful questions and valuable suggestions. We will improve the clarity and presentation of Section 3 accordingly.

---

### Official Review · Reviewer_KedN · 2025-03-14

**Overall Recommendation:** 1

**Summary:**

This paper investigates the phenomenon of "overthinking" in recent "o1-like" large language models (LLMs), where models expend excessive computational resources (generating many tokens and solution steps) on simple problems with minimal benefit to accuracy or reasoning diversity. The authors provide the first comprehensive analysis of this issue, introducing novel outcome and process efficiency metrics to quantify the inefficiency.

# update after rebuttal

Since most of my concerns remain and significant revisions are still needed, I strongly suggest another round of revision. Thanks.

**Claims And Evidence:**

Fine

**Essential References Not Discussed:**

Fine

**Experimental Designs Or Analyses:**

Fine

**Methods And Evaluation Criteria:**

Fine

**Other Comments Or Suggestions:**

Typo:
1. Figure 1 is referred to as "Figure 1(a)" in the text (line 041), but the figure itself doesn't have an "(a)" label.

**Other Strengths And Weaknesses:**

Hope the feedback is valuable for the authors and helps improve the quality in the revision or camera ready. I am happy to update my score after rebuttal if necessary. Thanks!

Pros:
1. Addresses a pertinent and relatively unexplored inefficiency ("overthinking") emerging in the latest generation of powerful reasoning LLMs.
2. Effectively demonstrates and quantifies the overthinking issue through clear examples (Fig 1, Fig 2) and quantitative analysis (solution distributions, token usage vs. difficulty).
3. Proposes novel "Outcome Efficiency" (ξo) and "Process Efficiency" (ξp) metrics that provide a more nuanced evaluation of LLM reasoning beyond simple accuracy, considering resource usage and solution diversity.
4. Develops and evaluates concrete self-training based methods (preference optimization + response simplification) that demonstrably improve efficiency without significant accuracy loss.

Cons:
1. It is suggested that the authors provide the code for reproduction.
2. The central concept relies on "o1-like" models, but this term isn't precisely defined beyond citing specific models (QwQ, DeepSeek-R1, Kimi). The paper could benefit from explicitly stating the defining characteristics of this model class (e.g., explicit multi-solution generation, specific internal reflection mechanisms cited by the model providers) that lead to the observed overthinking, rather than just relying on the "o1" name recognition.
3. It is not very clear how the authors set the hyperparameters such as temperatures. It suggested that the authors provide more details about all the experiment setting. Although temperature is briefly mentioned in "We generated 10 samples for each instance in the training dataset with a temperature of 1.0" (line 312), more explanations about the selection of the temperature and the temperature setting for the other parts of the study are desired.
4. It is not very clear about the impact of hyperparameters such as temperatures on the "Overthinking" issues. It is suggested that the authors conduct a more detailed analysis on the sensitivity of the hyperparameters.
5. Table 4 is hard to interpret. "+SimPOFCS+Reflection (Ours) 92.8 1.9 1330.7 80.0% 89.5%" does not have much difference compared to "+SimPOFirst-Correct Solution 91.0 1.4 1016.0 88.7% 98.1%" and "+SimPOGreedily Diverse Solutions 91.8 1.7 1286.1 84.3% 93.6%". It is not sure whether the proposed method has a big advantage.
6. The insight is limited and the overall observations are superficial. Why Some LLMs have the "Overthinking" issues? Do all reasoning LLMs have the "Overthinking" issues? What are the fundamental cause of the "Overthinking" issues? Does the proposed method solve the fundamental cause of the "Overthinking" issues?
7. There is no formal definition of the "Overthinking" issue. It is suggested the authors provide a formal definition of the "Overthinking" issue. Otherwise, it is not very clear.
8. The proposed solution seems incremental. It is mostly based on SimPO. It seems that the proposed method does not solve the fundamental cause of the "Overthinking" issues.

**Questions For Authors:**

See Weaknesses

**Relation To Broader Scientific Literature:**

Fine

**Theoretical Claims:**

Fine

---

> ### Author Rebuttal · Authors · 2025-04-01
>
> Thank you very much for your insightful questions and suggestions.
>
> Q1: Provide the code for reproduction.
>
> A1: We will make our code publicly available soon.
>
> ---
> Q2: The term "o1-like models" isn't precisely defined.
>
> A2: Thank you for pointing this out. We acknowledge that our original description of “o1-like” models lacks precision. A clearer term would be “Long Reasoning Models”, which refers to models that generate detailed CoT reasoning by iteratively producing intermediate reasoning steps and sequentially refining solutions until reaching a final conclusion.
>
> ---
> Q3&Q4:More details about the temperatures.
>
> A3&A4: We set the temperature to 1.0 for generating training data to promote diversity. For benchmark evaluations, we used a temperature of 0 to ensure reproducibility.
>
> We followed your suggestion and conducted a more detailed analysis regarding the impact of temperature settings. The table below summarizes the results of QwQ-32B-Preview evaluated on the Math Level 1 and 5. Clearly, the temperature hyperparameter has only a marginal effect on the number of generated solutions, further confirming our assertion that "overthinking" is a fundamental behavior inherent to these models.
>
> |   Level |   temperature |   solution_num |
> |--------:|--------------:|---------------:|
> |       1 |           0   |            3.7 |
> |       1 |           0.3 |            3.9 |
> |       1 |           0.5 |            3.6 |
> |       1 |           0.7 |            3.8 |
> |       1 |           1   |            3.6 |
> |||
> |       5 |           0   |            4   |
> |       5 |           0.3 |            3.9 |
> |       5 |           0.5 |            3.9 |
> |       5 |           0.7 |            4   |
> |       5 |           1   |            3.9 |
>
> ---
> Q5: Table 4 is hard to interpret. It is not sure whether the proposed method has a big advantage.
>
> A5: Both approaches demonstrate advantages in simplifying long CoT processes and mitigating overthinking. Nonetheless, considering benchmark performances and their suitability for challenging problems, we prefer "SimPO+FCS+Reflection" as our final recommended approach.
>
> ---
> Q6:The insight is limited and the overall observations are superficial.
>
> A6: Our results indicate that "overthinking" mainly happens in Long Reasoning Models, which learn advanced reasoning methods (e.g., reflection and backtracking) through reinforcement learning. When they fail to stop reasoning at the right time, they start overthinking.
>
> We confirmed our findings by evaluating two additional models, DeepSeek-R1 and QwQ-32B, which were released after our original submission, further proving that our results are robust. One important observation was that the formal (full-release) versions of the models provided higher accuracy than their preview versions, but exhibited much stronger overthinking behaviors. These results further demonstrate that the overthinking problem is a fundamental behavior inherent to these models.
>
> |Model|Acc|#Solutions|Length|
> |---|---|---|---|
> |R1-Preview|93.4|2.8|2168.6|
> |R1|96.4|4.3|2704.3|
> |QwQ-Preview|93.0|3.2|2407.9|
> |QwQ|96.4|7.1|4799.4|
>
> However, more in-depth research is still needed to fully understand the root causes of overthinking. While our current method effectively reduces this phenomenon, it doesn't completely solve the underlying problem. We plan to investigate more fundamental solutions in future work.
>
> Finally, it is worth noting that our primary contribution is presenting the first comprehensive study  that clearly explains the overthinking issue. We show that Long Reasoning Models frequently waste computational effort by generating unnecessary or repeated reasoning steps that add very little benefit. This notable contribution has been recognized positively by the other reviewers.
>
> ---
> Q7: There is no formal definition of the "Overthinking" issue.
>
> A7: Formally speaking, we define "overthinking" as a phenomenon where a long reasoning model repeatedly evaluates and revisits solutions within the CoT reasoning process, rather than converging promptly to a final answer. We will incorporate this formal definition explicitly in the Introduction section of our forthcoming manuscript revision.
>
> ---
> Q8: The proposed solution seems incremental.
>
> A8: We acknowledge that our proposed method does not completely resolve the fundamental cause of the "overthinking" phenomenon. Our primary contribution instead lies in providing the community’s first systematic analysis and explicit characterization of the overthinking problem, introducing quantitative metrics to assess efficiency specifically in Long Reasoning Models, and proposing viable mitigation strategies. We aim for our findings and benchmarks to provide a valuable framework to enhance the community’s overall understanding of these models' behavior and guide future research toward addressing overthinking more comprehensively.

---

### Decision · Program_Chairs · 2025-05-01

**Decision:**

Accept (poster)

**Comment:**

The paper tackles the overthinking problem in long reasoning models or o1 like reasoning models. It was found that the models generate unnecessarily long and repetitive reasoning chains for simple problems. The authors measures this issue and propose self-training methods to reduce the redundant computations without hurting accuracy. The work focuses on a practical problem and the experiment results are significant. But some reviewers felt that the analysis lacks depth and the evaluation and the evaluation is only limited to math tasks. There is definitely room for improvement but the paper could be interesting to some ICML audiences.